# Relationship between Leaf Traits and PM-Capturing Capacity of Major Urban-Greening Species

**Sanghee Park [1], Jong Kyu Lee [1], Myeong Ja Kwak [1], Yea Ji Lim [1], Handong Kim [1], Su Gyeong Jeong [1], Joung-a Son [2], Chang-Young Oh [2], Sun Mi Je [2], Hanna Chang [2], Kyongha Kim [1] and Su Young Woo [1],***

[1] Department of Environmental Horticulture, University of Seoul, Seoul 02504, Korea
[2] Urban Forests Research Center, National Institute of Forest Science, Seoul 02455, Korea
* Correspondence: wsy@uos.ac.kr; Tel.: +82-10-3802-5242

**Abstract:** High concentrations of airborne particulate matter (PM) in urban areas are of great concern to human health. Urban greening has been shown to be an effective and eco-friendly way to alleviate particle pollution, and attention to its role in mitigating particle pollution has increased worldwide. The species-specific PM-capturing capacity of ten urban-greening species in Seoul was evaluated by leaf functional traits (average leaf area (ALA), specific leaf area (SLA), and leaf width-to-length ratio (W/L)), microstructures (roughness, stomata, and trichomes), and physicochemical traits (contact angle ($\theta_w$), surface free energy ($r_s$), the work of adhesion for water ($W_a$), and epicuticular wax loads (EWL)). The relationships between leaf traits and PM adsorption by leaves were revealed by Pearson's correlations and principal component analysis (PCA). A gravimetric method was used to quantify, by particle size, the PM adsorbed on leaf surfaces or embedded in leaf epicuticular wax layers. The key factors for PM adsorption on leaf surfaces were the SLA, the mean roughness value ($R_a$), and stomatal size. The SLA and $R_a$ of adaxial leaf surfaces were negatively correlated with PM accumulation on leaf surfaces, while stomatal length and width were positively correlated with surface PM load. The $r_s$ and EWL positively affected the in-wax PM load. Species-specific PM deposition was the result of complicated mechanisms of various leaf traits. Three evergreen shrub species, *Buxus sinica* (Rehder & E.H. Wilson) M.Cheng var. *insularis* (Nakai) M.Cheng, *Taxus cuspidata* Siebold & Zucc., and *Euonymus japonicus* Thunb., were efficient in capturing both surface PM and in-wax PM. The PCA revealed that the high PM accumulation efficiency of these three species might be attributable to the interaction between stomatal size and EWL. *Aesculus turbinata* Blume, *Chionanthus retusus* Lindl. & Paxton, and *Rhododendron schlippenbachii* Maxim. had intermediate PM adsorption ability, which might be a result of interactions among stomatal density, the $W_a$ of adaxial surfaces, and ALA. *Magnolia denudata* Desr., *Styphnolobium japonicum* (L.) Schott, *Liriodendron tulipifera* L., and *Ginkgo biloba* L. had low PM accumulation efficiency. These four species exhibited correlations among SLA, the $R_a$ of adaxial leaf surfaces, and W/L, which had negative effects on PM adsorption.

**Keywords:** urban greening; PM capturing ability; leaf trait; microstructure; physicochemical traits

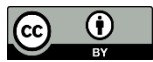

## 1. Introduction

Rapid urbanization and industrialization have resulted in particle pollution in the recent Anthropocene epoch [1,2]. Particulate matter (PM) is a generic term for a mixture of fine solid particles and liquid droplets found in the atmosphere. Particulate matter can be subdivided according to aerodynamic diameter ($D_a$) into $PM_{10}$ ($D_a$, <10 μm) and $PM_{2.5}$ ($D_a$, <2.5 μm). Particulate matter results not only from natural emissions (such as wildfires, volcanic eruptions, and sea salt formation) but also from anthropogenic processes, such as fuel combustion and vehicle emissions [3,4]. Particulate matter can negatively affect human health, especially the cardiopulmonary system, and has therefore become a major

issue worldwide [3]. The effects on human health vary with the size and chemical composition of the PM [5]. The World Health Organization (WHO) estimated that approximately 3 million deaths and 85 million DALYs (disability-adjusted life years, which account for both the length and quality of life) were attributable to particle pollution in 2012 [6].

Particles are scavenged from the air when they impact a surface after any of the following five processes: (1) sedimentation, (2) diffusion, (3) turbulence, (4) washout, and (5) occult deposition [7]. Wet deposition is a mechanism for the removal of particles by washout through precipitation (rainfall or snow) or, to a lesser extent, the impact of fog or cloud droplets on vegetation [8]. Dry deposition occurs via processes such as sedimentation, turbulence, and Brownian motion. Particles with $D_a$ smaller than 1 μm diffuse across concentration gradients and behave like gaseous pollutants because of Brownian motion [7,9]. Other than directly regulating the PM source, urban greening is the most effective measure to reduce air pollutants once PM has been emitted into the air [10]. It is universally recognized that urban vegetation offers a sustainable path for improving the air quality in polluted areas [7,11]. Trees can scavenge particles and gaseous pollutants such as $NO_2$, $SO_2$, $O_3$, and CO [12]. Specifically, leaves as the major sinks of trees can capture PM from the air [13,14]; PM can be adsorbed on leaf surfaces or embedded in the wax layers of leaves [15]. Previous studies have reported the potential of plants' ability to purify the air and that the service of urban trees has become crucial.

The quantitative analysis of PM is strongly affected by different environmental conditions and plant species, so it is crucial during scientific investigations to examine and control the factors influencing the PM-capturing capacity of different tree species [16]. Many studies have assessed the efficiency of PM capture across different plant species and identified relationships between PM deposition on leaves and leaf morphological variability at both the micro and macro levels [16–20]. Sæbø et al. [17] examined PM accumulation in 22 trees and 25 shrubs in Norway and Poland, and there were 10–15 times higher PM amounts on Taxus and Pinus belonging to conifer species. Chen et al. [21] reported that the groove area ratio and trichome density are important leaf traits for high-efficiency $PM_{2.5}$ capture. Sgrigna et al. [16] analyzed the relationship between leaf traits and the amount of PM on leaf surfaces sampled from twelve tree species in Italy and presented an accumulation index by scoring each leaf trait that significantly influenced PM adsorption. Leaf traits such as trichome density and leaf surface roughness are positively correlated with PM deposition on leaf surfaces [3,21,22]. In contrast, the specific leaf area and contact angle are negatively correlated with the quantity of the surface PM adsorption [3,17,23–26]. In addition, the structure, amount, and composition of wax can influence leaf surface wettability [8,27,28] and, thus, the accumulation of PM [15,29,30].

Particle pollution is a serious public health problem in Seoul. In 2016, the concentration of $PM_{2.5}$ in Seoul was 26 μg/m³, which is about twice as high as the WHO air quality criterion (10 μg/m³) and that of major cities in 2015 (Tokyo 13.8 μg/m³, London 11 μg/m³, and LA 14 μg/m³). According to a report of $PM_{2.5}$ watches and warnings by the Korea Ministry of Environment in 2021, there were 128 watches and 1 warning in 2017, which increased to 315 watches and 1 warning in 2018 and 590 watches and 52 warnings in 2019. Urban greenspaces can assist in ameliorating particle pollution, but urban and suburban areas have limited available open space [17,31]. Therefore, there is a need to quantify the PM-capturing ability of different taxa, identify species with high PM-capturing ability, and optimize the configuration of green spaces.

Although several studies have discussed the leaf traits associated with PM deposition, few studies have been conducted on the major urban species in Seoul. Moreover, there remain limitations to understanding the mechanisms of the PM deposition of plant species. The hypothesis of this study was that several leaf traits vary among the major urban species, which affects their species-specific PM-capturing capacity. Hence, this work focused on (1) investigating the relationship between foliar traits and PM adsorption on leaves and (2) quantifying the PM-capturing ability of ten major urban tree species in Seoul.

## 2. Materials and Methods

### 2.1. Study Site and Plant Material

The study site was Seoul Forest Park, located in Seongdong-gu, Seoul, South Korea. The average $PM_{10}$ and $PM_{2.5}$ concentrations (43.6 µg/cm$^3$ and 24.4 µg/cm$^3$, respectively) that were measured at a monitoring station (about 700 m away from a sampling site) in Seoul Forest Park over the past five years were similar to the nationwide concentrations (Table S1). Ten major urban greening species in Seoul were selected for this study: *Aesculus turbinata* Blume, *Chionanthus retusus* Lindl. & Paxton, *Ginkgo biloba* L., *Liriodendron tulipifera* L., *Magnolia denudata* Desr., *Styphnolobium japonicum* (L.) Schott, *Taxus cuspidata* Siebold & Zucc., *Buxus sinica* (Rehder & E.H. Wilson) M.Cheng var. *insularis* (Nakai) M.Cheng, *Euonymus japonicus* Thunb., and *Rhododendron schlippenbachii* Maxim. (Figure S1). The main characteristics of the selected species are presented in Table S2.

### 2.2. Sampling

The location of each plant was marked during a preliminary investigation for each species when leaves from identical trees were collected by consecutive sampling (Figure 1). The average height (m) and diameter at the breast height (DBH, cm) of the sampled trees are shown in Table S3. Branches were sampled from five individuals of each species. For each tree, about four branches in different directions were harvested at 3–6 m (tree species) height or 0.5–1 m (shrub species) above ground level on the following dates: 23 July, 13 August, 9 September, 9 October, and 30 October. After sampling, the samples were immediately placed in plastic bags to prevent contamination, transported to the laboratory, and stored in a refrigerator at 4 °C before analysis.

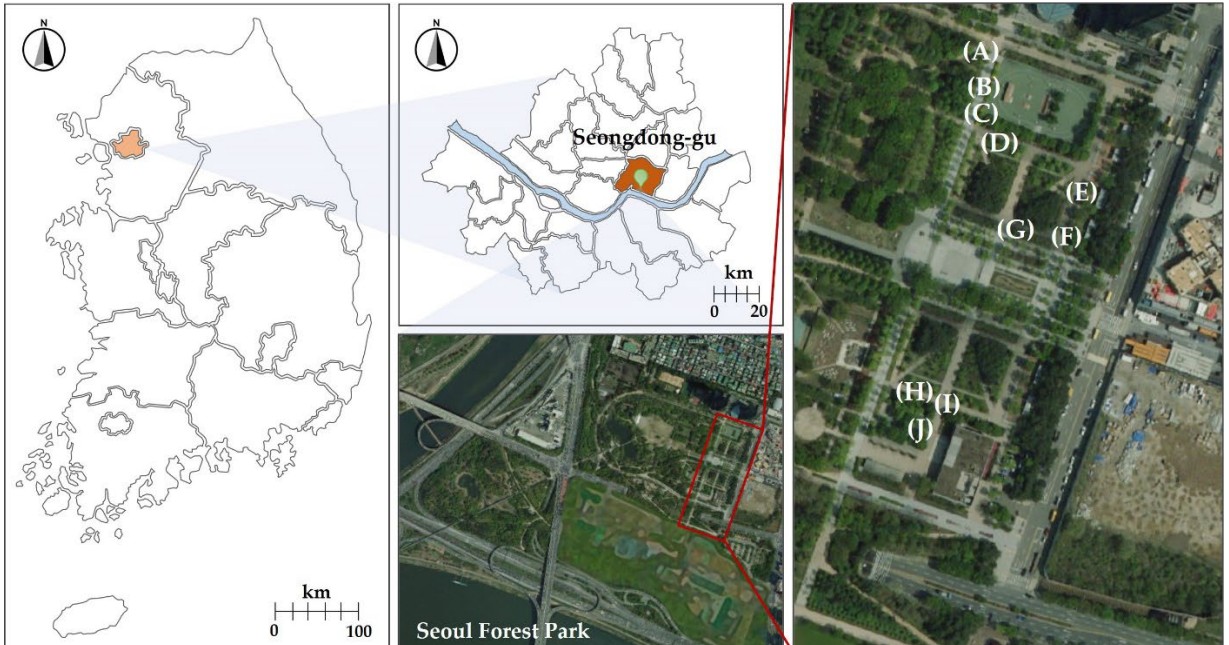

**Figure 1.** Location of each sample point for each species in Seoul Forest Park. *L. tulipifera* (A); *G. biloba* (B); *E. japonicus* (C); *M. denudata* (D); *B. koreana* (E); *R. schlippenbachii* (F); *S. japonicum* (G); *C. retusus* (H); *T. cuspidata* (I); *A. turbinata* (J). Source of satellite image: National Geographic Information Institute of MOLIT (The Ministry of Land, Infrastructure, Transport), the Republic of Korea (http://map.ngii.go.kr/ms/map/Aerial.do, accessed on 23 December 2021).

### 2.3. Leaf Functional Traits

A specified number of leaves were detached from the sampled branches using plant-trimming scissors. The leaf length, width, and area were measured using WinFOLIA image software (WinFOLIA, Regent Instruments Inc., Quebec, QC, Canada); the leaves were

then dried in an oven for forty-eight hours at 60 °C. Dried leaves of each batch were weighed using an electronic balance with an accuracy of 0.001 g. Average leaf area (ALA), specific leaf area (SLA), and leaf width-to-length ratio (W/L) were calculated according to [3] and [20].

### 2.4. Leaf Micromorphological Traits

After being separated from the sampled branches, the leaves were placed in a Petri dish and freeze-dried using a lyophilizer (FD 8508, IlShinBioBase Co. Ltd., Dongducheon, Korea). Leaf samples from broad-leaved species were cut to a size of 5.0 mm × 5.0 mm with plant-trimming scissors. For *T. cuspidata*, 0.5 mm-long needles were excised. Four leaf sections, two from each adaxial and abaxial surface, were cut, attached to a metal stub with conductive double-sided carbon tape, and platinum-coated using an ion sputter coater (MC1000, Hitachi, Tokyo, Japan). Field-emission scanning electron microscopy (FE-SEM; SU8010, Hitachi High-Tech, Tokyo, Japan) was used to observe the microstructures of the leaf surfaces.

The parameters related to leaf surface roughness were examined using a non-contact surface profiler (Bruker Contour GT-K™, Bruker Nano GmbH, Berlin, Germany). The non-contact surface profiler uses optics (white light interferometry and interference microscopy) instead of a stylus that can impose contact pressure on a specimen [32,33]. The $R_a$ (the arithmetic mean deviation from the average line within the assessed length) and $R_t$ (the total of the maximum peak height and the maximum valley depth of roughness) were measured to grade the roughness of adaxial and abaxial leaf surfaces.

### 2.5. Leaf Physicochemical Traits

The contact angle ($\theta$) is the angle between the solid surface and the line tangent to the liquid droplet passing through the contact point between the surface and the droplet. Leaf wettability is determined by measuring the contact angle between a water droplet and the leaf surface [34]. A larger spherical water droplet is produced as a result of the lower hydrophilicity of the leaf surface, which also causes a lower contact angle. The static contact angle can be determined using the sessile drop method [35]. The leaf samples were cut with a razor blade to avoid the midrib and mounted on a glass microscope slide. The contact angles were measured using two test liquids: deionized water and diiodomethane [36]. A 10 μL droplet of distilled water or 2 μL of diiodomethane at room temperature (25 °C) was gently placed on adaxial and abaxial leaf samples using a pipette.

Measurement of the contact angle of a test liquid on the surface, with known surface tension and parameters relevant to surface tension (dispersive and polar components), is a common method for obtaining the surface free energy of the solid surface ($r_s$). The $r_s$ is a parameter that affects surface-related characteristics, such as adsorption, wetting, and adhesion [37]. When the liquid contacts the solid surface in the gaseous phase, the relationship between the solid and liquid interfacial tension at the triple line between the liquid, solid, and gas can be expressed by Young's equation in the absence of the spreading pressure [36,38]:

$$r_{sl} = r_s - r_l \cos \theta \tag{1}$$

where $r_{sl}$, $r_s$, and $r_l$ represent the interfacial tension between the solid and liquid phases, the surface free energy of the solid and liquid (mJ/m$^2$), respectively, and $\theta$ (°) is the contact angle at the triple line between the solid, liquid, and gas. The surface free energy can be calculated by adding the dispersive and polar components [39] as follows:

$$r = r_d + r_p \tag{2}$$

where $r$, $r_d$, and $r_p$ are the surface free energy and the dispersive and polar components (mJ/m$^2$), respectively. The interfacial tension between a test liquid and a solid can be calculated using the geometric mean equation according to Owens and Wendt [40]:

$$r_{sl} = r_l + r_s - 2\sqrt{r_s^d r_l^d} - 2\sqrt{r_s^p r_l^p} \tag{3}$$

where $r_s{}^d$ and $r_l{}^d$ are the dispersive components of the solid and liquid (mJ/m$^2$), respectively; and $r_s{}^p$ and $r_l{}^p$ are the polar components of the solid and liquid (mJ/m$^2$), respectively. Combining Equations (1) and (3) yields:

$$r_l(1 + \cos\theta) = 2(\sqrt{r_l^p r_s^p} + \sqrt{r_l^d r_s^d} \tag{4}$$

The work of adhesion was defined as the reversible thermodynamic work needed for separating the interface from the equilibrium state of two phases to a separation distance of infinity. The work of adhesion for water ($W_a$) can be calculated following the Young–Dupré equation [36]:

$$W_a = r_l + r_s - r_{sl} = r_l(1 + \cos\theta) \tag{5}$$

Epicuticular wax loads (EWL) were quantified using colorimetric analysis [41,42]. This method is based on the color change caused by the reaction of wax with acidic potassium dichromate ($K_2Cr_2O_7$). The reagent was prepared by mixing 5 g of powdered potassium dichromate ($K_2Cr_2O_7$) with 10 mL of deionized water. The solution was mixed with 250 mL of concentrated sulfuric acid and heated under a fume hood until the slurry became clear. For the extraction of epicuticular wax, 30 pieces of leaves (total area of 30.16 cm$^2$) were prepared for each sample. In the case of the broad-leaved species, each piece was punched at a 0.8 cm diameter. For *T. cuspidata*, 30 entire leaves were used since the average leaf area investigated was 1.1 cm$^2$. The leaf samples were immersed in 10 mL of chloroform for 40 s at room temperature (~25 °C).

The extract was transferred to a 2 mL glass vial and evaporated at 70 °C for 30 min using a heating block (MaXtable H10, Daihan Scientific Co. Ltd., Seoul, Korea). After oxidation by adding 300 μL of potassium dichromate solution, the extract was heated for 30 min at 100 °C in a heating block. After heating, the glass vials were placed on ice for cooling, and 700 μL of deionized water was added to each vial. The samples were maintained at room temperature for 1 h for color development. The absorbance of each sample was measured at 590 nm using a microplate reader (Epoch microplate spectrophotometer, Synergy-Bio Tek, Winooski, VT, USA). Carnauba wax was used as the standard wax to develop a calibration curve. A calibration curve was used to calculate the EWL for each species. The EWL was expressed in mg/g.

### 2.6. Quantitative Assessment of PM

The quantitative assessment of PM was examined gravimetrically according to [43] (Figure S2). For each species, a number of leaves were randomly selected from the samples and were carefully detached using plant-trimming scissors. The leaves were placed in a glass container containing 250 mL of deionized water. The glass container was agitated for one minute using an ultrasonic bath (JAC-Ultrasonic 1505P) to remove particles deposited on the leaf surfaces. These represent water-soluble PM from the leaf surfaces (SPM). The leaf area of each sample was measured using WinFOLIA image software (WinFOLIA, Regent Instruments Inc., Quebec, QC, Canada) after rinsing with deionized water. The leaves were placed in 150 mL of chloroform in a glass container for 40 s to dissolve the epicuticular wax and separate PM stabilized in the wax layers. These represent the PM retained in leaf waxes (termed WPM). Deionized water and chloroform were stored in a refrigerator until filtration was performed.

To prevent electrostatic charge, the filters were passed through a deionizer gate and pre-weighed using an electronic balance with an accuracy of 0.001 mg. The washing solutions containing SPM were filtered sequentially through three types of filters with different retentions: nylon filter (pores 10 μm) and MCE (mixed cellulose ester) filter (pores 3

μm and 0.22 μm) (all MF-Millipore, Merck, Darmstadt, Germany). In the case of chloroform containing WPM, the filtration process was performed as for SPM, with the exception that filters were replaced with polytetrafluoroethylene PTFE membrane filters (Advantec, Tokyo, Japan) for organic solvents. The filtration procedure was performed using a filtration set with a 47 mm glass filter funnel connected to a vacuum pump. After filtration, the filters were dried (25 °C, RH 35%) for twenty-four hours and reweighed. Consequently, three PM sizes were left on each filter: (i) >10 μm, (ii) 3–10 μm, and (iii) 0.2–3 μm. The quantity of PM accumulated on the leaves or in wax layers for each species was determined as follows:

$$P \ (\mu g/cm) = (P_2 - P_1)/A \qquad (6)$$

where $P$ is the amount of leaf PM retention (μg/cm), $P_1$ is the pre-weighed amount of filter paper (μg), $P_2$ is the weight of filter paper with PM (μg), and A is the surface area of the leaves (cm²). Amounts of $PM_{10}$ were calculated as the total mass of $PM_{3-10}$ and $PM_{0.2-3}$. The leaf area was calculated using a double-sided surface because the particles were washed from both the adaxial and abaxial surfaces of the leaves.

*2.7. Leaf Area Index (LAI)*

The leaf area index (LAI) is defined as half the total area of leaves and woody materials per unit of ground surface area [44]. Hemispherical photography is a useful indirect method for estimating the LAI of forest canopies by providing a circular hemispheric projection. Using the LAI-2000 plant canopy analyzer (Li-cor Inc., Lincoln, NE, USA), hemispherical photography analyzes LAI by measuring the canopy gap fraction from different zenith angles and the angle measured directly above [45]. The LAI was estimated by taking digital hemispherical photographs (DHPs) of the canopy using a camera fitted with a fish-eye lens (a lens with a 180° circular view).

DHPs of the canopy of each species were taken from 1.2 m above ground using a smartphone camera with a 180° fish-eye lens fixed to a tripod with a leveling head, which ensured a horizontal position. The LAI was determined from the DHPs of the canopy using the Gap Light Analyzer (GLA) software. The PM load based on LAI (μg/cm) was quantified by multiplying the LAI value by the amount of PM adsorbed on leaf surfaces or embedded in the wax layers of ten species. It was hypothesized that the rate of PM adsorption would be identical in all parts of the plant.

*2.8. Statistical Analysis*

All data were statistically analyzed using SPSS Statistics 26 (SPSS Inc., IBM Company Headquarters, Chicago, IL, USA). The Shapiro–Wilk test was used to confirm the normality of all data. A one-way analysis of variance (ANOVA) as a parametric test was performed for the variables that satisfied normality, to determine whether there were statistically significant differences among different species, followed by Tukey's post hoc test. The non-parametric Kruskal–Wallis test was used for the variables that did not meet normality, to identify statistically significant differences among species. Pearson's correlation test was performed to test whether there is a statistically significant linear relationship between PM load and various foliar traits. Principal component analysis (PCA) was applied to investigate associations of specific leaf characteristics with amounts of accumulated particles on leaves.

## 3. Results

*3.1. Leaf Functional Traits*

Statistically significant differences in ALA, SLA, and W/L (ANOVA, $p = 0.000$) were observed among the ten species (Table 1). The ALA varied between $1.1 \pm 0.1$ cm² and $255.4 \pm 34.4$ cm². ALA was obviously higher in *A. turbinata* (255.4 cm²), followed by *L. tulipifera*,

*M. denudata*, *C. retusus*, *G. biloba*, *S. japonicum*, *E. japonicus*, and *R. schlippenbachii*. The smallest ALA groups were observed in *B. koreana* (1.4 cm²) and *T. cuspidata* (1.1 cm²). The SLA and the W/L ratio also varied by species. *L. tulipifera* and *S. japonicum* with thinner leaves exhibited larger SLAs, while *T. cuspidata* and *B. koreana* showed smaller SLA values by their thick and leathery leaves. The W/L ratio which reflects the leaf shape was higher in *G. biloba* (fan-shaped leaves) and *L. tulipifera* (broad ovate-shaped leaves) at 1.46 and 1.29, respectively. The W/L ratio of *S. japonicum* and *T. cuspidata* was close to zero because of the lanceolate or needle-shaped leaf.

**Table 1.** Average leaf area (ALA), specific leaf area (SLA), and leaf width-to-length ratio (W/L ratio) of tested plant species.

| Species | ALA | SLA | W/L Ratio |
|---|---|---|---|
| *Aesculus turbinata* | 225.4 ± 34.4 [a] | 169.9 ± 26.1 [b] | 0.42 ± 0.01 [ef] |
| *Chionanthus retusus* | 82.3 ± 5.4 [c] | 132.7 ± 5.9 [cd] | 0.58 ± 0.08 [cd] |
| *Ginkgo biloba* | 45.1 ± 3.8 [d] | 143.2 ± 7.1 [bcd] | 1.46 ± 0.10 [a] |
| *Liriodendron tulipifera* | 161.5 ± 22.7 [b] | 228.5 ± 15.2 [a] | 1.29 ± 0.01 [b] |
| *Magnolia denudata* | 147.2 ± 16.1 [b] | 145.6 ± 11.3 [bcd] | 0.67 ± 0.06 [c] |
| *Styphnolobium japonicum* | 19.6 ± 1.9 [de] | 228.7 ± 18.4 [a] | 0.33 ± 0.01 [f] |
| *Taxus cuspidata* | 1.1 ± 0.1 [e] | 57.9 ± 6.9 [e] | 0.17 ± 0.02 [g] |
| *Buxus koreana* | 1.4 ± 0.1 [e] | 55.5 ± 1.7 [e] | 0.48 ± 0.04 [de] |
| *Euonymus japonicus* | 19.1 ± 1.7 [de] | 118.8 ± 10.8 [d] | 0.51 ± 0.02 [de] |
| *Rhododendron schlippenbachii* | 16.4 ± 2.5 [de] | 153.9 ± 6.3 [bc] | 0.45 ± 0.05 [e] |

The same lowercase letter indicates that there were no statistically significant differences among species at a significance level of 0.05 (*n* = 5).

### 3.2. Leaf Micromorphological Traits

As summarized in Table 2, leaf micro-morphological traits differed among species. The mean stomatal density (No. mm⁻²) varied among the species by more than five-fold. The stomatal density was highest for *A. turbinata* (439 ± 44 mm⁻²), followed by *E. japonicus*, *M. denudata*, *B. koreana*, *R. schlippenbachii*, *T. cuspidata*, *S. japonicum*, *C. retusus*, *L. tulipifera*, and *G. biloba*. The stomatal size (μm), which is the length and width of the stomata on abaxial leaf surfaces, also varied among species. In the present study, a higher stomatal size was observed in *T. cuspidata* (42.3 μm long and 33.5 μm wide) and *B. koreana* (42.1 μm long and 32.4 μm wide). *S. japonicum* and *A. turbinata* showed lower stomatal sizes at 12.4 long and 6.2 μm wide and 12.3 μm long and 4.9 μm wide, respectively.

**Table 2.** Leaf micro-morphological traits (stomata and trichomes) of ten plant species.

| Species | Stomata | | | Trichome | | | |
|---|---|---|---|---|---|---|---|
| | Density | Size (μm) | | Types [z] | | Size (μm) [y] | |
| | (No. mm⁻²) | Length | Width | Adaxial | Abaxial | Adaxial | Abaxial |
| *A. turbinata* | 439 ± 44 | 12.3 ± 3.1 | 4.9 ± 1.8 | | NG | | 176 ± 64 |
| *C. retusus* | 157 ± 16 | 19.4 ± 2.4 | 7.5 ± 1.1 | PG | NG, PG | | 191 ± 48 |
| *G. biloba* | 84 ± 21 | 22.3 ± 3.6 | 14.8 ± 2.8 | | | | |
| *L. tulipifera* | 135 ± 8 | 19.9 ± 5.3 | 7.6 ± 2.1 | | | | |
| *M. denudata* | 208 ± 2 | 19.4 ± 3.2 | 6.4 ± 1.3 | NG | NG | 215 ± 28 | 411 ± 79 |
| *S. japonicum* | 161 ± 20 | 12.4 ± 1.7 | 6.2 ± 1.6 | NG, SG | NG | 202 ± 31 | 436 ± 111 |
| *T. cuspidata* | 163 ± 34 | 42.3 ± 3.3 | 33.5 ± 3.6 | | | | |
| *B. koreana* | 175 ± 32 | 42.1 ± 2.5 | 32.4 ± 2.4 | NG | | 50 ± 6 | |
| *E. japonicus* | 229 ± 16 | 26.7 ± 1.3 | 20.6 ± 0.9 | | | | |
| *R. schlippenbachii* | 174 ± 23 | 17.0 ± 1.7 | 8.4 ± 0.7 | NG | NG | 1336 ± 482 | 1355 ± 252 |

The same lowercase letter indicates that there were no statistically significant differences among species at a significance level of 0.05. [z]: NG, non-glandular; PG, peltate glandular; S, stellate. [y]: Length of non-glandular trichomes.

There were differences among the ten species in trichome type and size on adaxial and abaxial leaf surfaces. *G. biloba*, *L. tulipifera*, *T. cuspidata*, and *E. japonicus* had no trichomes on either surface of the leaves, while the remaining six species exhibited different trichome types on their leaf surfaces (Table 2). In particular, *C. retusus* was clearly distinguished from the other species by markedly peltate glandular trichomes on its adaxial surfaces and non-glandular and peltate glandular trichomes on its abaxial surfaces. *S. japonicum* showed the non-glandular and stellate trichomes on the adaxial leaf surfaces and the non-glandular trichomes on the abaxial leaf surfaces. *R. schlippenbachii* formed the longest trichomes on both its adaxial and abaxial leaf surfaces, while *B. koreana* formed the shortest trichomes only on the adaxial leaf surfaces.

$R_a$ and $R_t$ differed significantly among the ten species (ANOVA, $p$ = 0.000; Figure 2). The Ra of adaxial leaf surfaces were highest for *S. japonicum* and *G. biloba*, followed by *R. schlippenbachii*, *E. japonicus*, *L. tulipifera*, *A. turbinata*, *T. cuspidata*, *B. koreana*, *M. denudata*, and *C. retusus*. The $R_t$ of adaxial leaf surfaces were highest for *S. japonicum* and *R. schlippenbachii* and lowest for *C. retusus*. *G. biloba* was observed to have higher roughness on the abaxial surfaces by the greatest $R_a$. *C. retusus* had the lowest $R_a$ on its abaxial surfaces. The $R_t$ of the abaxial surfaces was highest for *A. turbinata*, followed by *S. japonicum*, *G. biloba*, and *M. denudata*. The $R_t$ of the abaxial surfaces was significantly lower in *T. cuspidata*, *R. schlippenbachii*, *L. tulipifera*, *E. japonicus*, *B. koreana*, and *C. retusus*. As shown in the 3D interferometry images (Figures S5 and S6), the roughness of the abaxial surfaces was greater than that of the adaxial surfaces. The 3D topography was reflected in the SEM micrograph.

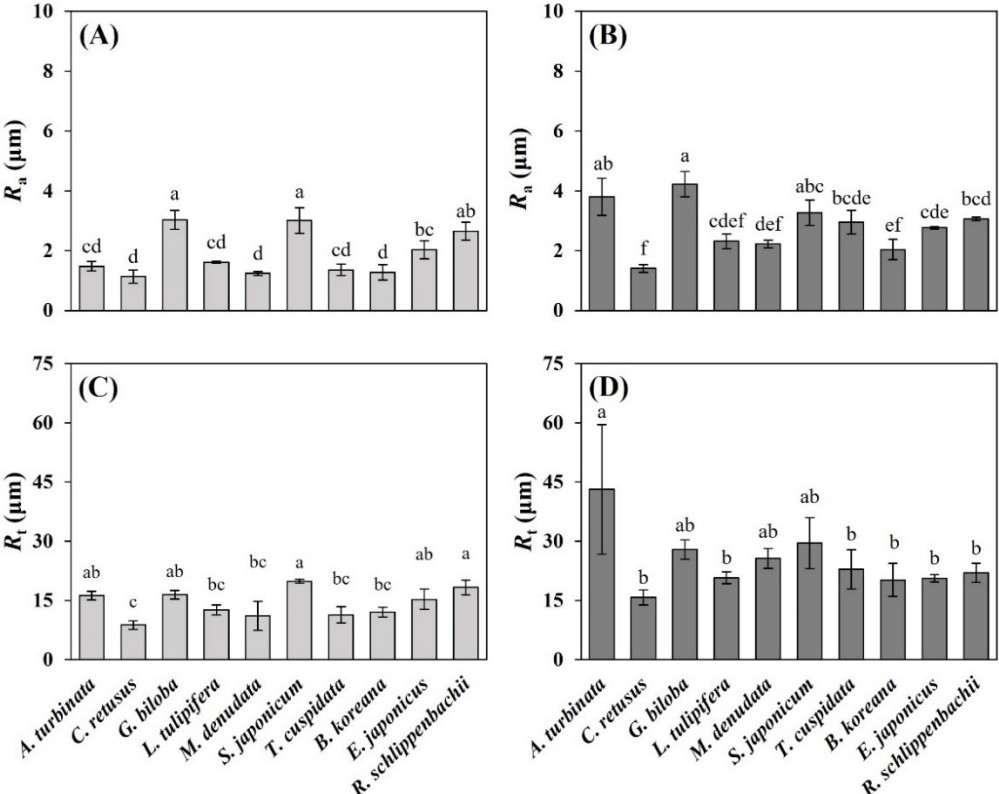

**Figure 2.** Roughness parameters ($R_a$ and $R_t$) of ten plant species. Adaxial surfaces (**A,C**); abaxial surfaces (**B,D**). Each column and error bar represents the mean and standard deviation, respectively ($n$ = 3). The same lowercase means there are no statistically significant differences among species at a significance level of 0.05.

There were obvious differences in the surface roughness of the different leaves among ten plant species (Figures S3 and S4). The ridges and grooves were densely distributed on both the adaxial and abaxial surfaces of *A. turbinata*, and most of these grooves

were quite narrow. *G. biloba* had well-developed grooves and ridges on both the adaxial and abaxial surfaces. *M. denudata* had a wide and shallow sunken area on the adaxial surfaces. *C. retusus* had a narrower sunken area than *M. denudata* on its adaxial surfaces and was covered with quite dense ridges and grooves around the stomata similar to *A. turbinata* (Figure S3). *S. japonicum* was densely covered with epicuticular wax crystals on the abaxial leaf surfaces. *L. tulipifera*, *B. koreana*, and *E. japonicus* had relatively smooth adaxial surfaces compared with the other species. In addition, *T. cuspidata* exhibited the presence of quite narrow and deep grooves on the abaxial surfaces (Figure S4).

### 3.3. Leaf Physicochemical Traits

The contact angles of the standardized water droplets on a leaf surface can be used as a proxy for leaf wettability. The water-repellent properties of leaf surfaces were applied to a high contact angle ($\theta$) of water droplets. The leaves were classified as "super-hydrophilic" surfaces if the $\theta$ was less than 40°, "highly wettable" if the $\theta$ was 40° to <90°, and "wettable" if the $\theta$ was 90° to <110°. If the $\theta$ was 110° to <130°, the leaves were termed as non-wettable and as highly non-wettable if the $\theta$ was >130° [46].

There were clear differences in the contact angles of deionized water and diiodomethane ($\theta_w$ and $\theta_d$, respectively) among ten plant species (ANOVA, $p = 0.000$; Table 3 and Figure S7). In the case of deionized water, the adaxial surfaces of *T. cuspidata* had the highest $\theta_w$ (114.1°), showing the most water-repellency. In contrast, *A. turbinata* had highly wettable adaxial surfaces, with the lowest $\theta_w$ values among the species. The abaxial surfaces exhibited higher $\theta_w$ values than the adaxial surfaces in most species experimented on, except for *R. schlippenbachii*. The abaxial surfaces of *A. turbinata* had the highest $\theta_w$ value (151.6°), showing a highly non-wettable surface, followed by *G. biloba* (143.6°). The $\theta_w$ values of *R. schlippenbachii* on the abaxial surfaces were the lowest among the ten species (73.7°). Overall, the $\theta$ values of diiodomethane were lower than those of deionized water for both the adaxial and abaxial leaf surfaces (Table 3)

**Table 3.** Contact angle of water ($\theta_w$) and diiodomethane ($\theta_d$) on the adaxial and abaxial leaf surfaces of ten plant species.

| Species | $\theta_w$ (°) | | $\theta_d$ (°) | |
|---|---|---|---|---|
| | **Adaxial** | **Abaxial** | **Adaxial** | **Abaxial** |
| *Aesculus turbinata* | 75.2 ± 3.2 [e] | 151.6 ± 1.5 [a] | 57.3 ± 1.0 [bc] | 69.8 ± 8.0 [bc] |
| *Chionanthus retusus* | 84.2 ± 4.7 [de] | 102.1 ± 9.4 [g] | 52.5 ± 3.0 [cd] | 59.6 ± 2.9 [cd] |
| *Ginkgo biloba* | 101.9 ± 3.5 [b] | 143.6 ± 4.7 [ab] | 64.8 ± 3.2 [ab] | 94.0 ± 2.3 [a] |
| *Liriodendron tulipifera* | 97.8 ± 5.4 [b] | 137.5 ± 3.6 [bc] | 57.3 ± 2.7 [bc] | 56.4 ± 5.3 [cd] |
| *Magnolia denudata* | 94.2 ± 1.0 [bcd] | 115.2 ± 1.5 [ef] | 69.1 ± 0.3 [a] | 69.3 ± 3.0 [bc] |
| *Styphnolobium japonicum* | 95.3 ± 4.2 [bc] | 132.1 ± 2.9 [cd] | 51.0 ± 5.8 [cd] | 81.9 ± 5.4 [ab] |
| *Taxus cuspidata* | 114.1 ± 9.9 [a] | 123.3 ± 2.6 [de] | 56.0 ± 7.6 [bc] | 82.1 ± 3.0 [ab] |
| *Buxus koreana* | 93.3 ± 1.0 [bcd] | 108.0 ± 3.4 [fg] | 59.2 ± 2.6 [bc] | 67.6 ± 9.9 [bc] |
| *Euonymus japonicus* | 84.5 ± 1.2 [cde] | 102.2 ± 6.4 [g] | 51.1 ± 1.2 [cd] | 67.6 ± 4.1 [bc] |
| *Rhododendron schlippenbachii* | 85.0 ± 3.7 [cde] | 73.7 ± 3.7 [h] | 43.4 ± 5.4 [d] | 51.9 ± 9.8 [d] |

The same lowercase letter indicates that there were no statistically significant differences among species at a significance level of 0.05 ($n = 4$).

The surface free energy varied among the ten species and between the adaxial and abaxial surfaces. As one of the thermodynamic properties of solid surfaces, the surface free energy ($r_s$) is associated with surface phenomena such as wettability and adhesion. There were significant differences in the $r_s$ of the adaxial and abaxial surfaces among the ten plant species (ANOVA, $p = 0.000$; Figure 3). In the species experimented on, the dispersion component dominated $r_s$ with a larger proportion than the polar components on both the adaxial and abaxial surfaces. The $r_s$ of the adaxial leaf surfaces descended in the following order: *R. schlippenbachii* (38.1 mJ/m$^2$), *T. cuspidata* (35.4 mJ/m$^2$), *A. turbinata* (34.5

mJ/m²), *E. japonicus* (34.3 mJ/m²), *C. retusus* (33.9 mJ/m²), *S. japonicum* (33.9 mJ/m²), *L. tulipifera* (30.4 mJ/m²), *B. koreana* (29.1 mJ/m²), *G. biloba* (26.0 mJ/m²), and *M. denudata* (23.9 mJ/m²). Unlike in other species, the polarity component accounted for a larger proportion of the $r_s$ on the adaxial surfaces of *A. turbinata*. The $r_s$ on the abaxial surfaces descended in the following order: *L. tulipifera* (47.0 mJ/m²), *A. turbinata* (40.2 mJ/m²), *R. schlippenbachii* (37.2 mJ/m²), *C. retusus* (29.8 mJ/m²), *B. koreana* (25.4 mJ/m²), *M. denudata* (25.3 mJ/m²), *E. japonicus* (24.7 mJ/m²), *S. japonicum* (21.3 mJ/m²), *T. cuspidata* (18.4 mJ/m²), and *G. biloba* (15.8 mJ/m²). The polarity component accounted for a larger proportion of the $r_s$ on the abaxial surfaces in *A. turbinata*, *L. tulipifera*, and *R. schlippenbachii*.

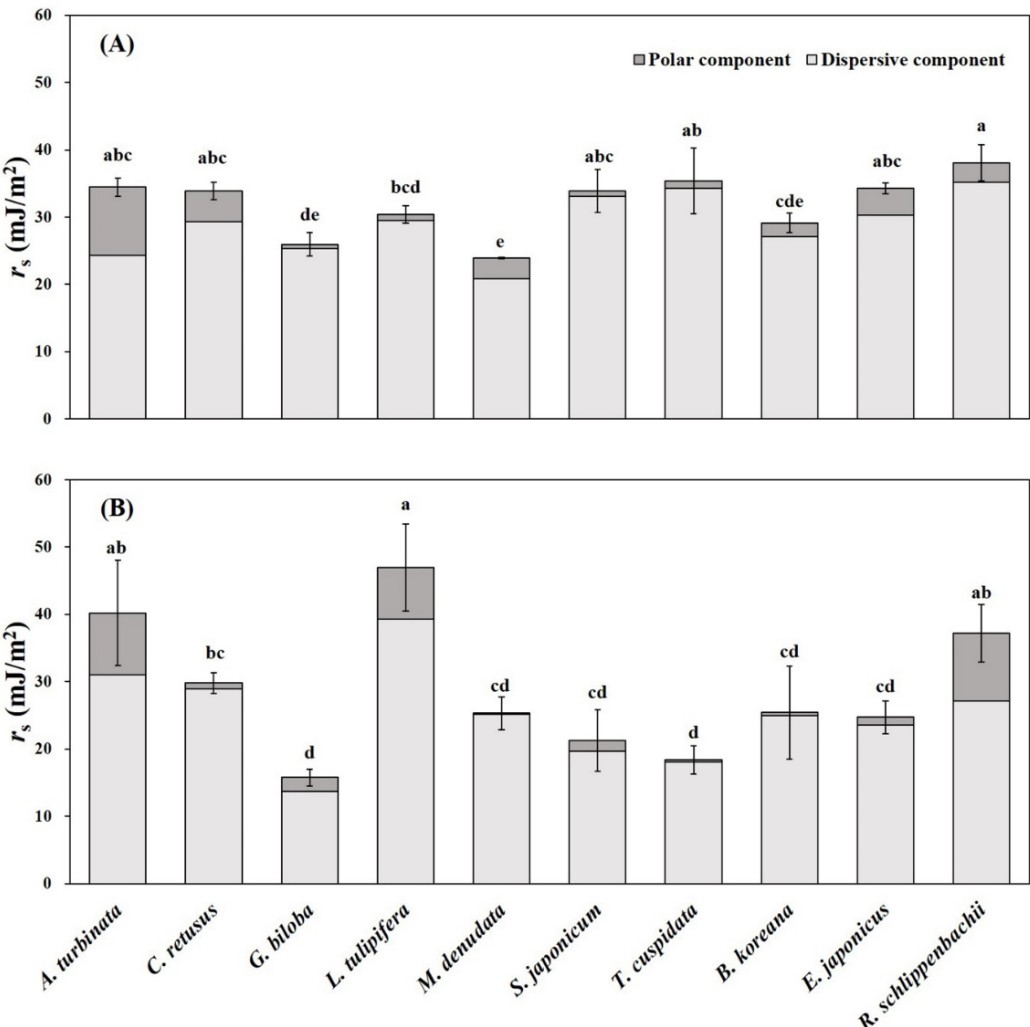

**Figure 3.** The surface free energy ($r_s$) on leaf adaxial surfaces (**A**) and abaxial surfaces (**B**) of ten species. The $r_s$ was expressed as the sum of the dispersive and polar components. Each column and error bar represents the mean and standard deviation, respectively (*n* = 4). The same lowercase letter indicates that there were no statistically significant differences among species at a significance level of 0.05.

The $W_a$ differed significantly among the ten species on both the leaf adaxial and abaxial surfaces (ANOVA, *p* = 0.000; Figure 4). The $W_a$ of the adaxial surfaces was greatest for *A. turbinata* (91.4 mJ/m²), followed by *C. retusus* (80.3 mJ/m²), *E. japonicus* (79.8 mJ/m²), *R. schlippenbachii* (79.2 mJ/m²), *B. koreana* (68.6 mJ/m²), *M. denudata* (67.5 mJ/m²), *S. japonicum* (66.1 mJ/m²), *L. tulipifera* (63.0 mJ/m²), and *G. biloba* (57.8 mJ/m²). The $W_a$ value of the adaxial surfaces was lowest for *T. cuspidata* (43.4 mJ/m²). The abaxial surfaces exhibited

higher $W_a$ values than those of the adaxial surfaces in most species, except for *R. schlippenbachii*. The $W_a$ of the abaxial surfaces varied more than 10-fold among the species. The $W_a$ of the abaxial surfaces was highest for *R. schlippenbachii* (93.2 mJ/m²), followed by *C. retusus* (57.3 mJ/m²), *E. japonicus* (57.5 mJ/m²), *B. koreana* (50.4 mJ/m²), *M. denudata* (41.9 mJ/m²), *T. cuspidata* (32.9 mJ/m²), *S. japonicum* (24.1 mJ/m²), *L. tulipifera* (19.2 mJ/m²), and *G. biloba* (14.4 mJ/m²). In contrast to the adaxial surface, the value for the abaxial surfaces was lowest for *A. turbinata* (8.78 mJ/m²).

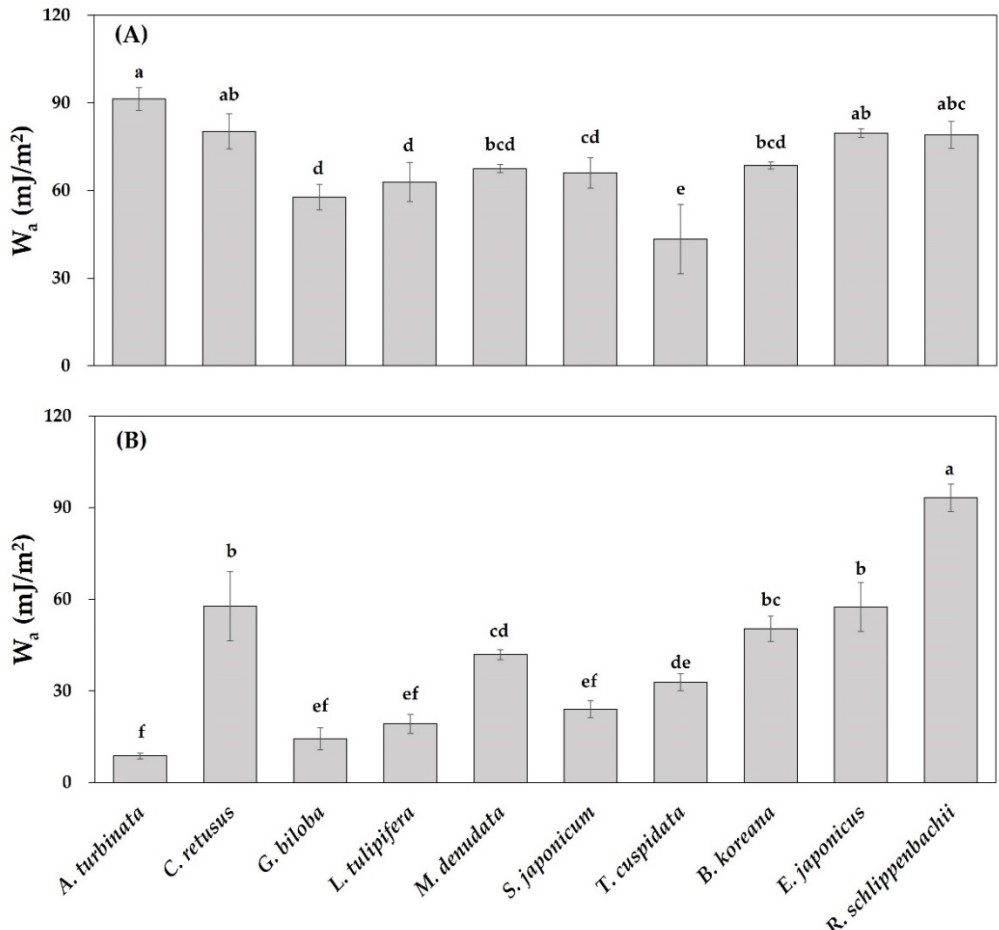

**Figure 4.** The work of adhesion for water (*W*a) of the ten species. Adaxial surfaces (**A**); abaxial surfaces (**B**). Each column and error bar represents the mean and standard deviation, respectively (*n* = 4). The same lowercase letter indicates that there were no statistically significant differences among species at a significance level of 0.05.

Significant differences were observed among the ten plant species in the epicuticular wax loads (EWL) (Kruskal–Wallis test, *p* = 0.001; Figure 5). *T. cuspidata* was identified as the species having the highest EWL, followed by *C. retusus*, *E. japonicus*, *S. japonicum*, *B. koreana*, *G. biloba*, *R. schlippenbachii*, and *L. tulipifera*. *A. turbinata* and *M. denudata* had lower EWLs.

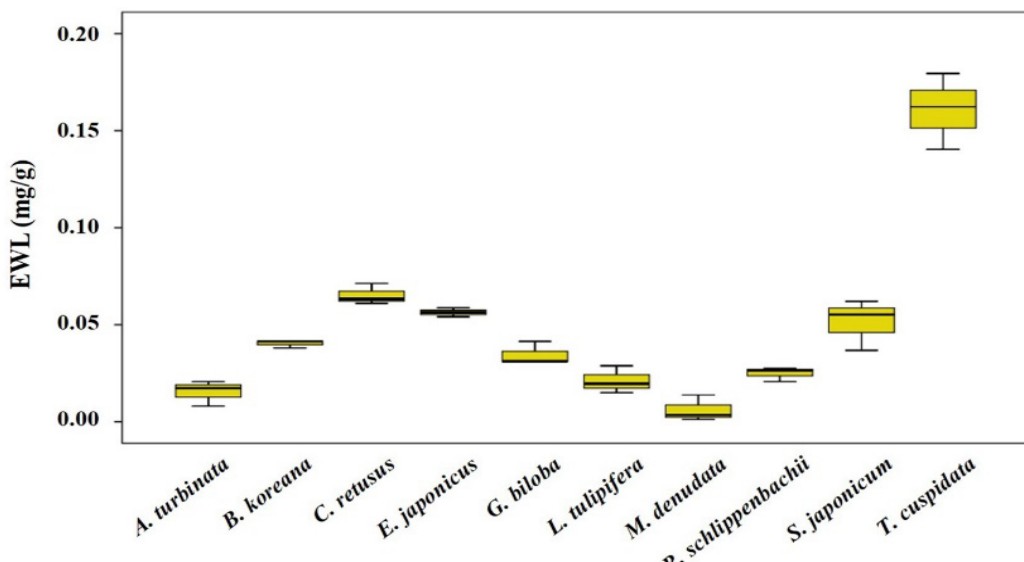

**Figure 5.** Box plots of the epicuticular wax loads (EWL) of ten plant species. Differences in wax quantity among ten species were tested using the non-parametric Kruskal–Wallis rank-sum test (*n* = 3 for each group). Box plots indicate median and interquartile ranges with bars representing maximum and minimum values. The significance level for homogeneous subsets was 0.05.

*3.4. PM Load on Leaf Surfaces and in Wax Layers*

The amounts of PM adsorbed on the leaf surfaces and accumulated in the wax layers differed significantly among the ten plant species (Kruskal–Wallis test, *p* = 0.000; Figure 6). $SPM_{10}$, $SPM_{2.5}$, $WPM_{10}$, and $WPM_{2.5}$ were analyzed for the ten species, taking the PM adsorption data from July to October as a whole.

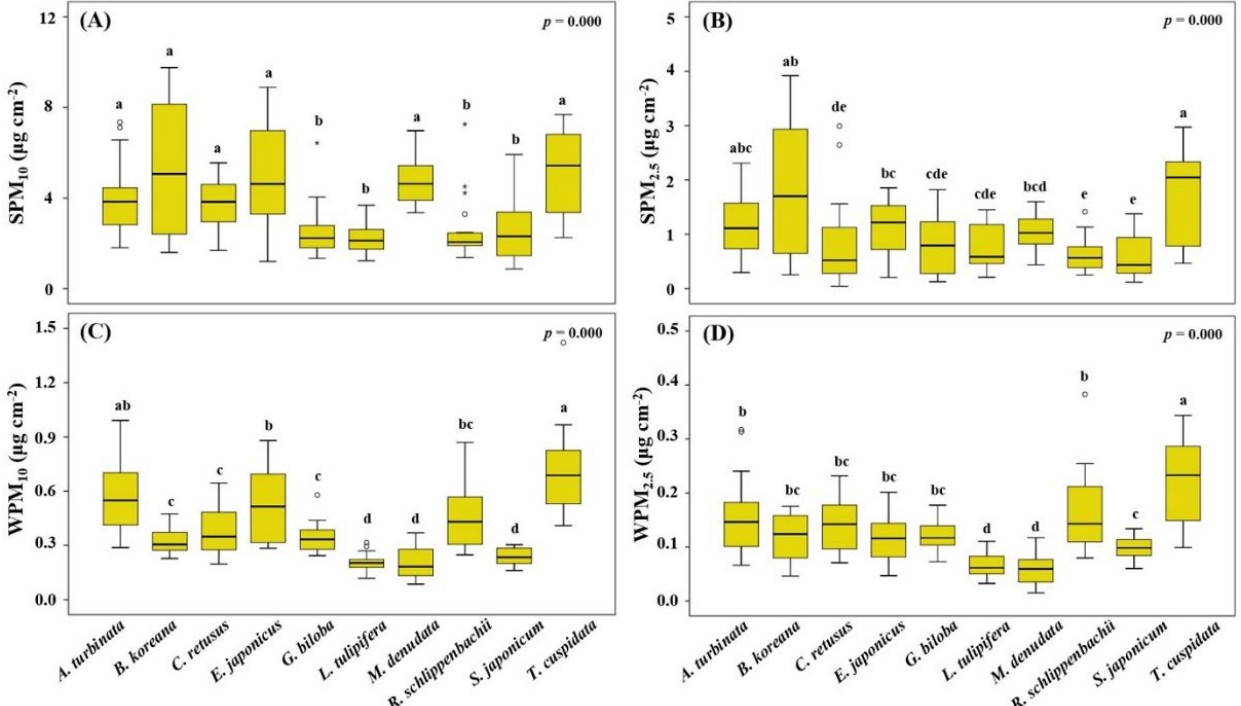

**Figure 6.** Box plots of the PM load on the leaves of the ten plant species. Surface $PM_{10}$ ($SPM_{10}$) (**A**); surface $PM_{2.5}$ ($SPM_{2.5}$) (**B**); wax $PM_{10}$ ($WPM_{10}$) (**C**); and wax $PM_{2.5}$ ($WPM_{2.5}$) (**D**). Differences in PM load among the species were tested using the non-parametric Kruskal–Wallis rank-sum test (*n* = 20 each group). Box plots with lowercase letters indicate the median and interquartile ranges with bars

representing the maximum and minimum values. The significance level for homogeneous subsets was 0.05.

The mean $SPM_{10}$ ranged from about 2.3 μg/cm² to 5.23 μg/cm². *B. koreana* had the highest $SPM_{10}$-capturing capacity (5.23 μg/cm²), followed by *T. cuspidata*, *E. japonicus*, *M. denudata*, *A. turbinata*, *C. retusus*, *S. japonicum*, *G. biloba*, *R. schlippenbachii*, and *L. tulipifera*. According to the Kruskal–Wallis test, *B. koreana*, *T. cuspidata*, and *A. turbinata* were in the same homogeneous subset (data not shown), showing high $SPM_{2.5}$ accumulations on their leaf surfaces. The following order was *E. japonicus*, *M. denudata*, *C. retusus*, *G. biloba*, *L. tulipifera*, *R. schlippenbachii*, and *S. japonicum*.

There were also significant differences in the $WPM_{10}$-capturing capacity among the species. The $WPM_{10}$ ranged from 0.20 μg/cm² to 0.71 μg/cm². *T. cuspidata* showed the highest $WPM_{10}$, followed by *A. turbinata*, *E. japonicus*, *R. schlippenbachii*, *C. retusus*, *G. biloba*, *B. koreana*, *S. japonicum*, *L. tulipifera*, and *M. denudata*. In regard to the $WPM_{2.5}$ accumulations, *T. cuspidata* was the only species in the high accumulation group according to the Kruskal–Wallis test ($0.22 \pm 0.08$ μg/cm²). *C. retusus*, *G. biloba*, *B. koreana*, *E. japonicus*, and *S. japonicum* had considerably lower accumulations of $WPM_{2.5}$. *L. tulipifera* and *M. denudata* had the lowest $WPM_{2.5}$-capturing capability (0.07 μg/cm² and 0.06 μg/cm², respectively). There were differences among the ten plant species in the ratio of the four class types to the total PM mass (Figure S8). In general, the mass of the $SPM_{10}$ dominated most of the total mass, accounting for 64–80% of the total PM load; $SPM_{2.5}$ accounted for 16–24%; $WPM_{10}$ accounted for 3–12%; and $WPM_{2.5}$ accounted for 1–5%.

### 3.5. Relationship between Leaf Traits and PM Capturing Capability

Pearson's correlation analysis was performed to identify the correlations between various foliar traits and the PM adsorbed on leaf surfaces or encapsulated in leaf epicuticular wax layers (Table 4). The $SPM_{10}$ was significantly negatively correlated with SLA and $R_a$ (adaxial surfaces) in leaf traits ($p < 0.01$). In contrast, the SL and SW showed significant positive correlations with $SPM_{10}$ ($p < 0.05$). The SLA, SL, and SW were highly significant factors correlated with $SPM_{2.5}$ ($p < 0.01$); the SLA was negatively correlated with $SPM_{2.5}$, whereas the SL and SW were positively correlated with $SPM_{2.5}$. The PM embedded in epicuticular wax layers showed different correlation results compared with the PM adsorbed on the leaf surfaces. The $WPM_{10}$ showed significant positive correlations with EWL. The $WPM_{2.5}$ showed significant positive correlations with EWLs and $r_s$ (adaxial surfaces).

**Table 4.** Correlations among PM load and foliar traits ($n = 10$).

| Leaf Traits | $SPM_{10}$ | | $SPM_{2.5}$ | | $WPM_{10}$ | | $WPM_{2.5}$ | |
|---|---|---|---|---|---|---|---|---|
| | r | p | r | P | r | p | r | p |
| ALA | −0.15 | 0.672 | −0.19 | 0.595 | −0.13 | 0.727 | −0.28 | 0.431 |
| SLA | −0.81 ** | 0.005 | −0.83 ** | 0.003 | −0.54 | 0.104 | −0.54 | 0.107 |
| W/L | −0.55 | 0.101 | −0.40 | 0.257 | −0.51 | 0.130 | −0.53 | 0.116 |
| SL | 0.69 * | 0.028 | 0.87 ** | 0.001 | 0.37 | 0.298 | 0.38 | 0.280 |
| SW | 0.67 * | 0.035 | 0.86 ** | 0.002 | 0.47 | 0.176 | 0.46 | 0.186 |
| SD | 0.29 | 0.415 | 0.18 | 0.612 | 0.41 | 0.243 | 0.18 | 0.619 |
| $R_a$ (ad) | −0.66 * | 0.039 | −0.59 | 0.075 | −0.16 | 0.659 | −0.07 | 0.841 |
| $R_t$ (ad) | −0.53 | 0.116 | −0.46 | 0.183 | −0.01 | 0.985 | −0.01 | 0.987 |
| $\theta_w$ (ad) | 0.08 | 0.817 | 0.30 | 0.395 | 0.00 | 0.995 | 0.13 | 0.718 |
| $r_s$ (ad) | −0.09 | 0.798 | −0.08 | 0.836 | 0.62 | 0.058 | 0.66 * | 0.037 |
| $W_a$ (ad) | −0.08 | 0.834 | −0.30 | 0.406 | 0.02 | 0.963 | −0.12 | 0.747 |
| $R_a$ (ab) | −0.34 | 0.332 | −0.18 | 0.616 | 0.28 | 0.425 | 0.23 | 0.523 |
| $R_t$ (ab) | −0.12 | 0.745 | −0.06 | 0.876 | 0.18 | 0.612 | 0.08 | 0.830 |
| $\theta_w$ (ab) | −0.17 | 0.642 | 0.03 | 0.934 | −0.08 | 0.829 | −0.16 | 0.649 |
| $r_s$ (ab) | −0.37 | 0.292 | −0.28 | 0.428 | −0.13 | 0.722 | −0.18 | 0.626 |

| | | | | | | | | |
|---|---|---|---|---|---|---|---|---|
| $W_a$ (ab) | 0.13 | 0.730 | −0.06 | 0.861 | 0.10 | 0.774 | 0.19 | 0.599 |
| EWL | 0.41 | 0.236 | 0.49 | 0.153 | 0.64 * | 0.047 | 0.71 * | 0.022 |

SPM₁₀, surface $PM_{10}$; SPM₂.₅, surface $PM_{2.5}$; WPM₁₀, in-wax $PM_{10}$; WPM₂.₅, in-wax $PM_{2.5}$; ALA, average leaf area; SLA, specific leaf area; W/L, leaf width-to-length ratio; SL, stomatal length; SW, stomatal width; SD, stomatal density; $R_a$, the arithmetic mean deviation from the average line within the assessed length; $R_t$, the sum total of the maximum peak height and the maximum valley depth of roughness; $\theta_w$, contact angle of water; $r_s$, surface free energy; $W_a$, adhesion for water; EWL, epicuticular wax load; ad, adaxial surface; ab, abaxial surface. We determined the statistical significance of the Pearson Correlation Coefficient to test whether there is a statistically significant linear relationship between PM load and various foliar traits. *P* values were indicated by * $p < 0.05$ and ** $p < 0.01$.

Subsequently, the PM loads for the two size fractions on the leaf surfaces and in the wax and the values indicating foliar traits were used as variables for principal component analysis (PCA). For $R_a$, $R_t$, $\theta_w$, $W_a$, and $r_s$, only the results of the leaf adaxial surfaces were used for the PCA analysis. As shown in Figure 7, the cumulative contribution rate of the first two of nine principal components (PCs) represented 64.38% of the total variance. The first PC, based on all four types of PM load, SL, SW, EWL, W/L, and SLA, accounted for 36.3% of the variance. In particular, PC1 was positively correlated with SL, SW, and EWL and negatively correlated with W/L and SLA. The second PC explained 28.1% of all factors, primarily based on the following characteristics: ALA, SD, $W_a$, $\theta_w$, and W/L. The second PC was positively correlated with ALA, SD, and $W_a$; however, it was negatively correlated with W/L and $\theta_w$. The amount of SPM and WPM was positively correlated with the following leaf traits: SW, SL, EWL, $r_s$, and SD, but it was negatively correlated with SLA, ALA, W/L, Ra, and Rt. The factor coordinates and scores of the first two PCs are shown in Figure 7. The ten tested species were divided into the following two groups by PC1: (1) *A. turbinata*, *C. retusus*, *T. cuspidata*, *B. koreana*, and *E. japonicus*, and (2) *G. biloba*, *L. tulipifera*, *M. denudata*, *S. japonicum*, and *R. schlippenbachii*.

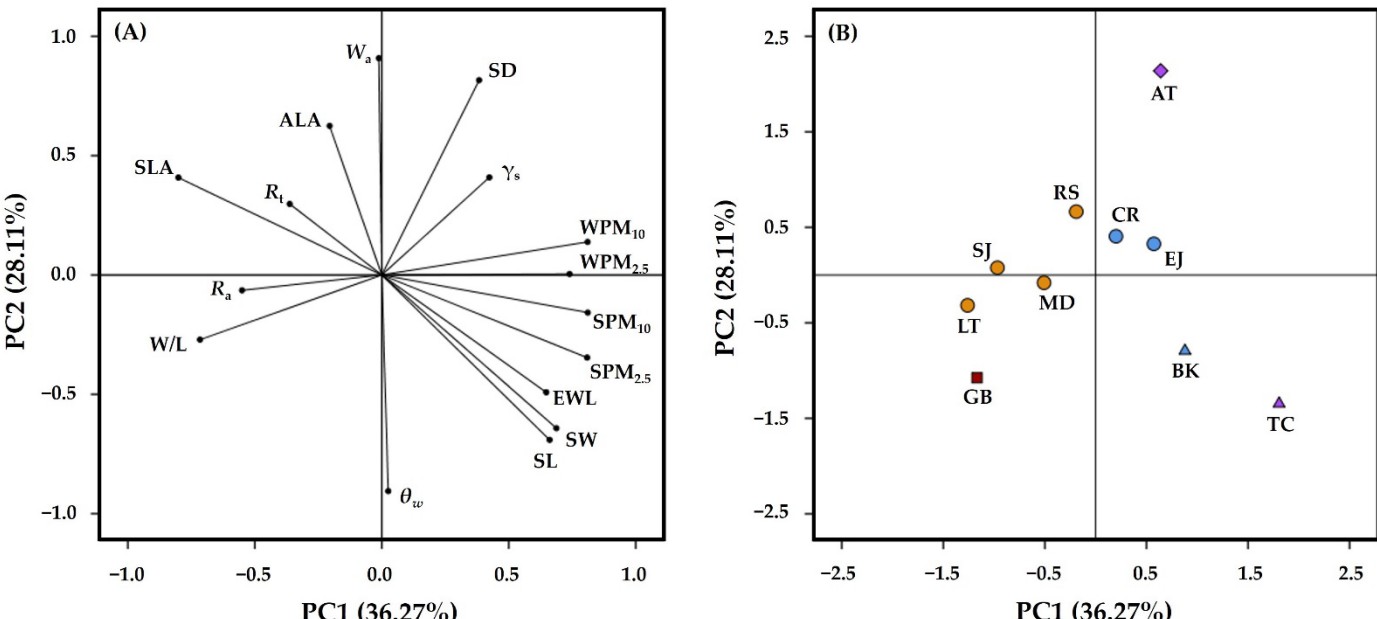

**Figure 7.** The biplot graph of the factor coordinates (**A**) of 16 variables and of factor scores (**B**) of the first two PCs. SPM₁₀, surface $PM_{10}$; SPM₂.₅, surface $PM_{2.5}$; WPM₁₀, in-wax $PM_{10}$; WPM₂.₅, in-wax $PM_{2.5}$; ALA, average leaf area; SLA, specific leaf area; WL, leaf width to length ratio; SL, stomatal length; SW, stomatal width; SD, stomatal density; $R_a$, the arithmetic mean deviation from the average line within the assessed length; $R_t$, the sum total of the maximum peak height and the maximum valley depth of roughness; $\theta_w$, contact angle of water; $r_s$, surface free energy; $W_a$, work of adhesion for water; EWL, epicuticular wax load, AT, *A. turbinata*; CR, *C. retusus*; GB, *G. biloba*; LT, *L. tulipifera*;

MD, *M. denudata*; SJ, *S. japonicum*; TC, *T. cuspidata*; BK, *B. koreana*; EJ, *E. japonicus*; RS, *R. schlippenbachii*.

## 4. Discussion

The foliar traits examined in this study exhibited a wide variation among the ten plant species, resulting in variable PM adsorption. There was a negative correlation between the SPM load and SLA (Table 4). Previous studies have reported that deposition velocity has a negative relationship with SLA. It is generally believed that leaves with low SLA might have thicker palisade tissues. Thus, leaves with low SLA can have structural stability, which may allow them to adsorb more particles to leaf surfaces [3,20]. Chiam et al. [3] reported that there was no significant correlation with PM deposition velocity, which is in line with the results of this study. In contrast to the results of the present study, Zhang et al. [20] reported that the PM deposition velocity was negatively correlated with W/L. The W/L indicated the extent of the proximity of the leaf edges to the main vein; a lower W/L might cause leaves to sway less, reducing the resuspension of PM.

Leaf microstructure is considered to be one of the major factors influencing PM deposition. The stomata can be one of the places where $PM_{2.5}$ or $PM_{10}$ is deposited on leaf surfaces [31]. The SPM was positively correlated with stomatal size, but not with stomatal density in this study (Table 4). Zha et al. [47] reported that stomatal size was significantly associated with the load of $PM_{2.5}$. Liang et al. [48] found no significant relationship between stomatal density (ranging averagely from 10.4 to 38.4 No./mm$^2$), and $PM_{2.5}$. In contrast, Simon et al. [49] reported that plant species with high stomatal density exhibited higher foliar PM accumulation. The species tested in that study had a relatively high average stomatal density, ranging from 237 to 757 No./mm$^2$. Chen et al. [21] confirmed that when the stomatal density is high, it might have a significant relationship with PM capture. In the present study, an average stomatal density of 84–439 No./mm$^2$ was observed (Table 2), showing no significant correlation with the PM load.

In the present study, the relationship between $R_a$ and the PM load differed from that reported in previous studies. In this study, $R_a$ was negatively correlated with $SPM_{10}$. Chen et al. [21] reported that the groove area ratio (a percent ratio of groove area to total leaf area) was positively correlated with $PM_{2.5}$ capture in broadleaved tree species. In addition, Zhang et al. [22] reported a positive correlation between leaf surface roughness and the adsorption of $PM_{2.5}$ in broadleaved species. In this study, $R_t$ did not show a statistically significant correlation with PM loads. Wang et al. [4] reported that *Ulmus pumila* L., which had 1 μm wide parallel ridges detected by using a non-contact optical profiler, had a more efficient PM-capturing capacity than *Ginkgo biloba* L., which had the greatest P–V distance (the distance between the highest peak and lowest valley, which is the same as $R_t$ in this study). In addition, the P–V distance did not have a significant relationship with PM adsorption. It seems that not only the roughness parameters related to the height of ridges but also the width between ridges should be considered to identify relationships with PM adsorption.

The work of adhesion is one of the physicochemical properties of a material, along with the surface free energy and its polar and dispersive components. By measuring the work of adhesion of a liquid, it is possible to quantify the degree of droplet retention or repellence on a plant surface [36,50]. Dry and wet deposition can be decreased by the low adhesion of water drops on leaf surfaces [51]. Wang et al. [36] reported that leaf water drop adhesion is positively correlated with $r_s$ ($r = 0.535$, $p = 0.000$) and $W_a$ ($r = 0.698$, $p = 0.000$). In the present study, the $r_s$ of leaf adaxial surfaces was positively correlated with WPM (Table 4). It is likely that a higher $r_s$ allowed greater water drop adhesion on the leaf surface,s which increased the dry and wet deposition of PM. Unlike $r_s$, $W_a$ had no significant effect on the PM load.

The leaves with a higher EWL had a greater adsorption capacity for WPM (Table 4). Popek et al. [15] found that the amount of wax differed significantly among species, and the in-wax $PM_{2.5–10}$ load was positively correlated with wax quantities. In *Corylus colurna*,

high positive correlations were found between wax quantity and total in-wax PM content ($r$ = 0.95). However, when all the tested species were analyzed together, a moderate positive correlation was observed between in-wax PM$_{2.5-10}$ load and wax quantities ($r$ = 0.54). Moreover, Popek et al. [15] confirmed that PM deposition can be differently affected by the complicated mechanisms of various leaf traits, not only wax load.

PM has the potential to adversely affect human health, particularly in urban areas. Trees capture PM and accumulate toxic components such as heavy metals in an environmentally friendly manner [52]. It is valuable to select species that can capture PM when planning green infrastructure. In this study, ten major urban plant species in Seoul were tested to investigate the differences in their PM-capturing capacity. Significant differences in the amount of PM adsorbed on leaf surfaces and encapsulated in wax layers were observed among the species. Figure 7 shows that the PM load was positively correlated with the size and density of stomata, $r_s$, and EWL, and negatively correlated with the functional leaf traits (SLA, ALA, and W/L) and surface roughness parameters ($R_a$ and $R_t$), which coincided with the results of Pearson's correlation analysis (Table 4).

The species-specific PM-capturing capacity of the ten plant species that were tested can be explained by interpreting the results of the PCA with respect to the different PM loads among the ten species (Figure 7). The species in the first group were distinguished by PCA as having a relatively higher PM-capturing capacity than the species in the second group. *Taxus cuspidata* and *B. koreana* had high PM accumulation both on leaf surfaces and in epicuticular wax layers. The first principal component indicated that PM load was positively correlated with stomatal size (SL and SW) and EWL, and negatively correlated with SLA and W/L. *Taxus cuspidata* had the greatest mean stomatal size (both SL and SW; Table 2) and mean EWL (Figure 6). In addition, *T. cuspidata* had the lowest values for both SLA and W/L (Table 1). The two species of *B. koreana* and *T. cuspidata* had the lowest mean SLA and the largest stomatal size (Tables 1 and 2).

The high PM-capturing capability of *T. cuspidata* and *B. koreana* is in line with the results of previous studies. He et al. [23] reported the highest load of both PM$_{10}$ and PM$_{2.5}$ over winter in another member of the *Taxus* genus, English yew (*Taxus baccata* L.), due to the rough leaf surface. Reference [17] also found that *T. baccata* is an efficient species for PM capture. Kwon et al. [53] reported that *B. koreana* deposited the greatest quantity of PM$_{2.5-10}$ in both sampling dates due to the round shape and relatively wide base of leaves which allow little movement by wind. Among the deciduous broadleaved species, *A. turbinata* had greater PM-capturing capability (Figure 6) and was classified by PC1 into the first group.

*Aesculus turbinata* has not been discussed in the literature to the best of our knowledge; however, several studies have investigated other species in the *Aesculus* genus, particularly the horse chestnut (*Aesculus hippocastanum* L.) [17,54]. Tomašević et al. [54] reported that *A. hippocastanum* had a higher density of fine particles, as compared with *C. colurna*. *Euonymus japonicus* also had relatively high PM accumulation. Zhang et al. [55] reported that *E. japonicus* had a relatively high PM-capturing capacity with the greatest coarse PM (PM$_{2.5-10}$) load on the leaf surfaces because it is a low-growing shrub species with proximity to the pollutant source.

The two species of *S. japonicum* and *L. tulipifera* were differentiated by PC1 as showing relatively low PM-capturing capacity, which coincides with the result of the PM load (Figure 6). These two species had the highest mean SLA values, in contrast to *T. cuspidata* and *B. koreana* (Table 1). Additionally, *G. biloba* showed intermediate efficiency in capturing PM (Figure 6), which was identified by the PCA technique as being rather low in *G. biloba* (Figure 7). Specifically, *Ginkgo biloba* had the highest mean value of W/L (Table 1), which is the factor showing a negative correlation with PC1 (Figure 7). Many previous studies have included *G. biloba* when testing for species differences in PM-capturing capability and have confirmed *G. biloba* as a species with low PM-capturing efficiency [20,26,55–57]. Kwak et al. [56] reported that among five temperate tree species *G. biloba* had the lowest PM adsorption per unit of leaf area. Zhang et al. [26] reported that *G. biloba*

adsorbed a small mass of PM on its leaf surfaces because of the smooth and hydrophobic surface. Three evergreen species, *B. koreana*, *E. japonicus*, and *T. cuspidata*, were distinguished from most of the other deciduous species, except *for A. turbinata* and *C. retusus*. Evergreen species have the potential to function as biological filters throughout the year, whereas deciduous species lose their leaves and PM-capturing ability during the winter season [21,23]. In addition, this study proved that the PM-scavenging ability of evergreen species was more efficient than that of deciduous species. Due to the toxic components in urban air-suspended PM particles, however, it is necessary to investigate the physiological traits of each species for a better understanding of the PM-capturing capacity of plant species. More importantly, Lhotská et al. [58] demonstrated that soil dust PM, which can lead to different physicochemical properties, toxicokinetics, and surface reflectances induced biochemical, physiological, and metabolic alterations that drive epigenetic changes as the degree of leaf damage caused by oxidative stress.

## 5. Conclusions

Urban greening is one of the sustainable ways to alleviate air quality in urban areas deteriorated by high PM emission loads. We demonstrated that there was a significant relationship in leaf functional, micromorphological, and physicochemical traits for PM adsorption among major urban species in Seoul. The combinations of various leaf traits seemed to affect different PM-capturing capabilities of ten plant species. Among the leaf functional traits and micro-morphological traits, SLA and stomatal size (length and width of stomata) had significant effects on SPM adsorption; SLA was negatively correlated with SPM load, while stomatal size had a positive correlation with SPM. In addition, the leaf physicochemical trait EWL was significantly positively correlated with WPM, PM load immobilized in waxes. Given the relationships between the leaf traits and PM capture efficiency of urban plant species, we suggest that evergreen species, *B. koreana*, *T. cuspidata*, and *E. japonicus* are the best urban species that can be planted in urban areas to reduce particle pollutants faster and better during the four seasons because of their relatively large stomatal size and high epicuticular wax loads. Future studies need to clarify those potential explanations of meteorological factors throughout seasonal changes in physiological traits and PM-capturing capacity.

**Supplementary Materials:** The following supporting information can be downloaded at https://www.mdpi.com/article/10.3390/horticulturae8111046/s1, Table S1: Average PM concentration (μg/m$^3$) over five years, 2016–2020; Table S2: Basic characteristics and leaf photos of the selected plant species in this study; Table S3: Diameter at breast height (DBH, cm) and height (m) of sampled trees in Seoul Forest Park, 2019–2020; Figure S1: Photos of the seven tree and three shrub species in Seoul Forest Park; Figure S2: Schematic diagram of quantitative analysis of PM adsorption; Figure S3: SEM images showing microstructures of adaxial and abaxial surface and leaf morphology of five species; Figure S4: SEM images showing microstructures of adaxial and abaxial surface and leaf morphology of five species; Figure S5: 3D interferometry images (50× magnification) of adaxial leaf surfaces of ten plant species; Figure S6: 3D interferometry images (50× magnification) of abaxial leaf surfaces of ten plant species; Figure S7: Images of shape of test liquids (deionized water and diiodomethane) on leaf adaxial and abaxial surfaces of each ten tested plant species; Figure S8: Percent contributions of four class types to the PM mass of ten species.

**Author Contributions:** Conceptualization, S.Y.W.; methodology, S.P., J.K.L. and M.J.K.; validation, J.-a.S., C.-Y.O., S.M.J. and H.C.; formal analysis, S.P; investigation, J.K.L., Y.J.L., H.K. and S.G.J.; resources, J.K.L., Y.J.L. and H.K.; data curation, S.P., J.K.L., Y.J.L., H.K. and S.G.J.; writing—original draft preparation, S.P.; writing—review and editing, M.J.K.; visualization, S.P. and M.J.K.; supervision, S.Y.W. and K.K.; project administration, J.K.L.; funding acquisition, J.-a.S., C.-Y.O., S.M.J. and H.C. All authors have read and agreed to the published version of the manuscript.

**Funding:** This study was carried out with the support of 'A Study on Mechanism and Function Improvement of Plants for Reducing Air Pollutants' (Grant No. FE0000-2018-01-2021) from the National Institute of Forest Science (NIFoS).

**Data Availability Statement:** The data presented in this study are available on request from the corresponding author. The data are not publicly available due to privacy or other restrictions.

**Acknowledgments:** Special thanks go to the Center for Research Facilities at the University of Seoul and the Joint Device Center at the National Arboretum Baekdudaegan under the Korea Forest Service for the microscope technical support in the experimental field.

**Conflicts of Interest:** The authors declare no conflicts of interest. The funders had no role in the design of the study; in the collection, analyses, or interpretation of data; in the writing of the manuscript, or in the decision to publish the results.

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
