# Peer review of "Relationship between Leaf Traits and PM-Capturing Capacity of Major Urban-Greening Species"

_horticulturae, doi:10.3390/horticulturae8111046_

Round 1

Reviewer 1 Report

This paper presents the relationship between Leaf Traits and Particulate matter (PM) of Major Urban-Greening species. The paper is written properly, with enough results presented. However, some revisions needed to improve the paper, such as:

1. The urgency of the proposed work is not clearly described.

2. Too many table results are written repeatedly in the paragraphs. The authors could summarize and describe the main finding of the results.

3. The first sentence in Section 5, Conclusion, should be omitted.

Author Response

Author Reply to the Review Report

(Manuscript ID: horticulturae-1996731)

Dear Editor,

Title

Relationship between Leaf Traits and PM-Capturing Capacity of Major Urban-Greening Species

We are pleased to resubmit for publication the revised version of Manuscript ID: horticulturae-1996731 entitled “Relationship between Leaf Traits and PM-Capturing Capacity of Major Urban-Greening Species”. We are very grateful to the Associate Editor and three Reviewers for their deep and detailed comments which have helped us to improve our manuscript. We have revised our manuscript according to the suggestions and comments of three anonymous reviewers. Grammatical error correction and English improvement were carried done by a native-speaking English editor as suggested. Please find below point-by-point responses to each of the reviewers’ comments on our manuscript. Upload files of revised manuscripts were attached in two forms: ‘Track Changes’ and ‘Final PDF version’. Finally, we hope that reviewers and editors will be satisfied with the response to reviewer comments.

Sincerely yours,

Response to Reviewer 1 Comments

This paper presents the relationship between Leaf Traits and Particulate matter (PM) of Major Urban-Greening species. The paper is written properly, with enough results presented. However, some revisions needed to improve the paper, such as:

Point 1: The urgency of the proposed work is not clearly described.

Response 1: (lines 12-14) We greatly appreciate Reviewer 1's comment that helped improve the manuscript. As requested, we have clearly described the proposed work in the ‘Abstract’ section.

Point 2: Too many table results are written repeatedly in the paragraphs. The authors could summarize and describe the main finding of the results.

Response 2: We thank the reviewer for bringing up this point. As requested, we concisely summarized and presented the main findings related to our research questions or hypotheses in the ‘Results’ section.

Point 3: The first sentence in Section 5, Conclusion, should be omitted.

Response 3: (line 597) According to the reviewer's suggestion, we deleted and revised the first sentence in the ‘Conclusion’ section.

Reviewer 2 Report

1.      In the Introduction section: it will be better if you also add information about urban species (trees and shrubs) and their leaf traits that associated with the PM-capturing ability from previous studies. And also, urban species (trees and shrubs) that already applied to reduce particle pollutants in the cities or polluted area.  

2.      In the Discussion section: please also add brief information about the mechanism of leaves in capturing particulate matter and correlated with the morphological, physiological, and molecular characteristics of leaves.

1.      In the Conclusions section: please add suggestion related to the best urban species (trees and shrubs) that can be planted to the urban area to reduce particle pollutants faster and better.  

Author Response

Author Reply to the Review Report

(Manuscript ID: horticulturae-1996731)

Dear Editor,

Title

Relationship between Leaf Traits and PM-Capturing Capacity of Major Urban-Greening Species

We are pleased to resubmit for publication the revised version of Manuscript ID: horticulturae-1996731 entitled “Relationship between Leaf Traits and PM-Capturing Capacity of Major Urban-Greening Species”. We are very grateful to the Associate Editor and three Reviewers for their deep and detailed comments which have helped us to improve our manuscript. We have revised our manuscript according to the suggestions and comments of three anonymous reviewers. Grammatical error correction and English improvement were carried done by a native-speaking English editor as suggested. Please find below point-by-point responses to each of the reviewers’ comments on our manuscript. Upload files of revised manuscripts were attached in two forms: ‘Track Changes’ and ‘Final PDF version’. Finally, we hope that reviewers and editors will be satisfied with the response to reviewer comments.

Sincerely yours,

Response to Reviewer 2 Comments

Point 1: In the Introduction section: it will be better if you also add information about urban species (trees and shrubs) and their leaf traits that associated with the PM-capturing ability from previous studies. And also, urban species (trees and shrubs) that already applied to reduce particle pollutants in the cities or polluted area.

Response 1: (lines 71-77) Based on Reviewer 2's comment, we presented more detailed information about leaf traits that are associated with the PM-capturing ability of urban species from the previous studies in the ‘Introduction’ section.

In the ‘Introduction’ section (lines 71-77)

Many studies have assessed the efficiency of PM capture across different plant species and identified relationships between PM deposition on leaves and leaf morphological variability at both the micro and macro levels [16-20]. Sæbø et al. [17] examined PM accumulation in 22 trees and 25 shrubs in Norway and Poland and there were 10–15 times higher PM amounts on Taxus and Pinus belonging to conifer species. Chen et al. [21] reported that the groove area ratio and trichome density are important leaf traits for high-efficiency PM2.5 capture. Sgrigna et al. [16] analyzed the relationship between leaf traits and the amount of PM on leaf surfaces sampled from twelve tree species in Italy and presented an accumulation index by scoring each leaf trait that significantly influenced PM adsorption.

Point 2: In the Discussion section: please also add brief information about the mechanism of leaves in capturing particulate matter and correlated with the morphological, physiological, and molecular characteristics of leaves.

Response 2: (lines 589-595) As requested, we added brief information about the mechanism between leaf physiological and molecular traits associated with PM capture capability of leaves in the ‘Discussion’ section.

In the ‘Discussion’ section (lines 589-595)

Due to the toxic components in urban air-suspended PM particles, however, it is necessary to investigate the physiological traits of each species for a better understanding of the PM-capturing capacity of plant species. More importantly, Lhotská et al. [58] demonstrated that soil dust PM, which can lead to different physicochemical properties, toxicokinetics, and surface reflectances, induced biochemical, physiological, and metabolic alterations that drive epigenetic changes as the degree of leaf damage caused by oxidative stress.

Point 3: In the Conclusions section: please add suggestion related to the best urban species (trees and shrubs) that can be planted to the urban area to reduce particle pollutants faster and better.

Response 3: (lines 606-610) According to the reviewer's suggestion, we presented related to the best urban species (trees and shrubs) that can be planted in the urban area to reduce particle pollutants faster and better in the ‘Conclusion’ section.

In the ‘Conclusion’ section (lines 606-610)

Given the relationships between leaf traits and PM capture efficiency of urban plant species, we suggest that evergreen species, B. koreana, T. cuspidata, and E. japonicus are the best urban species that can be planted in urban areas to reduce particle pollutants faster and better during the four seasons, because of their relatively large stomatal size and high epicuticular wax loads.

Reviewer 3 Report

The title „Relationship between Leaf Traits and PM-Capturing Capacity 2 of Major Urban-Greening Species” in my opinion is suggestive for the content of this study.

In the abstract, I think the aim of the work (purpose) is not very clearly identified. It could be specified more clearly in my opinion. I also think that the justification for the research should be specified more clearly and included in the abstract.

The work as a whole is complex and quite well organized. This fact shows and demonstrates the correct and detailed documentation of the authors on the chosen topic. The results are clearly presented.

I also recommend to make a short linguistic check.

The discussions and conclusions are thoroughly supported by the results presented in the paper, which are very well illustrated. The bibliography is also complex.

Author Response

Author Reply to the Review Report

(Manuscript ID: horticulturae-1996731)

Dear Editor,

Title

Relationship between Leaf Traits and PM-Capturing Capacity of Major Urban-Greening Species

We are pleased to resubmit for publication the revised version of Manuscript ID: horticulturae-1996731 entitled “Relationship between Leaf Traits and PM-Capturing Capacity of Major Urban-Greening Species”. We are very grateful to the Associate Editor and three Reviewers for their deep and detailed comments which have helped us to improve our manuscript. We have revised our manuscript according to the suggestions and comments of three anonymous reviewers. Grammatical error correction and English improvement were carried done by a native-speaking English editor as suggested. Please find below point-by-point responses to each of the reviewers’ comments on our manuscript. Upload files of revised manuscripts were attached in two forms: ‘Track Changes’ and ‘Final PDF version’. Finally, we hope that reviewers and editors will be satisfied with the response to reviewer comments.

Sincerely yours,

Response to Reviewer 3 Comments

Point 1: The title „Relationship between Leaf Traits and PM-Capturing Capacity of Major Urban-Greening Species” in my opinion is suggestive for the content of this study.

Response 1: We greatly appreciate Reviewer 3's comment that helped improve the manuscript.

Point 2: In the abstract, I think the aim of the work (purpose) is not very clearly identified. It could be specified more clearly in my opinion. I also think that the justification for the research should be specified more clearly and included in the abstract.

Response 2: (lines 12-15) As requested, we clearly indicated the specific objectives of the present study in the ‘Abstract’ section.

In the ‘Abstract’ section (lines 71-77)

High concentrations of airborne particulate matter (PM) in urban areas are of great concern to human health. Urban greening has been shown to be an effective and eco-friendly way to alleviate particle pollution, and attention to its role in mitigating particle pollution has increased worldwide.

Point 3: The work as a whole is complex and quite well organized. This fact shows and demonstrates the correct and detailed documentation of the authors on the chosen topic. The results are clearly presented.

Response 3: We greatly appreciate Reviewer 3's comment that helped improve the manuscript. We revised the manuscript and concisely summarized the main findings related to our research questions or hypotheses in the ‘Results’ section.

Point 4: I also recommend to make a short linguistic check.

Response 4: After undergoing English language editing, we revised our manuscript.

Point 5: The discussions and conclusions are thoroughly supported by the results presented in the paper, which are very well illustrated. The bibliography is also complex.

Response 5: We wrote the bibliography according to the reference format in Instructions for Authors of MDPI.
